# Assessment of Welfare Problems in Broilers: Focus on Musculoskeletal Problems Associated with Their Rapid Growth

**DOI:** 10.3390/ani14071116

**Published:** 2024-04-05

**Authors:** Byung-Yeon Kwon, Jina Park, Da-Hye Kim, Kyung-Woo Lee

**Affiliations:** Department of Animal Science and Technology, Konkuk University, 120 Neungdong-ro, Gwangjin-gu, Seoul 05029, Republic of Korea; byung64@konkuk.ac.kr (B.-Y.K.); jinaa97@konkuk.ac.kr (J.P.); kdh142536@naver.com (D.-H.K.)

**Keywords:** animal welfare, broiler, footpad dermatitis, hock burn, stocking density

## Abstract

**Simple Summary:**

Animal welfare has become a global concern in the poultry industry. Over the years, broilers have been selectively bred for rapid growth and efficiency, resulting in birds with high body weight with large proportion of breast meat. However, this selective breeding has also led to welfare problems. Since the 2000s, there has been growing interest in alternative production systems that highlight animal welfare and sustainability, including free-range and organic farming methods. In addition, studies have been conducted to provide reliable information to consumers by measuring animal welfare levels. How to accurately assess the state of animal welfare at the farm levels is a central issue in animal welfare research. This review summarized the factors affecting the welfare status of broiler chickens reared in commercial broiler farms.

**Abstract:**

The growth of the broiler industry has been accompanied with concerns over the environmental and social impacts on intensive production systems, as well as the welfare of the animals themselves. As a result, since the 2000s, there has been growing interest in alternative production systems that improve animal welfare and sustainability. In this context, it is important to prioritize the welfare of broilers in commercial production systems and to use reliable welfare indicators to provide consumers with information about the welfare of the animals they consume. Resource-based measures (RBM) are based on assessing the resources available to the birds in terms of their housing, environment, and management practices, such as stocking density, litter quality, lighting and air quality, etc. Outcome-based measures (OBM), also known as animal-based measures, focus on assessing the actual welfare outcomes for the birds, such as footpad dermatitis, hock burn, contamination or damage to feathers, gait score, mortality, etc. These OBM-based measures are one of the more direct indicators of welfare and can help identify any welfare issues. The present review highlighted the factors that affect animal welfare indicators focused on OBMs which can be used in the commercial broiler farms.

## 1. Introduction

Animal welfare is defined as how well an animal is doing and dealing with the way it lives. Animal welfare has become a global concern in the poultry industry. The broiler that has been bred for rapid growth, improved feed efficiency, and digestibility has resulted in welfare problems such as leg health, contact dermatitis, and heat stress [1]. Boiler chickens raised for market weight can develop leg deformations despite whether they are raised in good environmental conditions, highlighting the need for a more comprehensive approach to welfare [2].

How to assess the state of animal welfare on a farm is a central issue in animal welfare research [3]. Various assessment criteria to practically evaluate the state of animal welfare are needed to accommodate market demands [4]. Over the years, many studies and collaborative projects like the Welfare Quality Network and the AssureWel project have helped create and implement thorough tools for assessing the welfare of animals on farms, especially those raising livestock [5,6,7]. Various resource-based measures (RBMs) and outcome-based measures (OBMs) are able to provide effective indicators of animal welfare for chickens [8]. As far as we know, no extensive studies have been conducted to assess the welfare for broilers raised in commercial meat production systems. Thus, the use of animal- and/or resource-based welfare measures to assess the welfare of broilers on commercial broiler farms will be considered of importance. Although the welfare indicators for broilers are well described in RBMs or OBMs, environmental factors affecting those welfare indicators are not well studied. The present paper briefly summarizes and highlights the factors affecting welfare indicators in broiler chickens focused on OBMs. In addition, this will also look at more animal-based causes and impacts on wellbeing, which can help detect wellbeing issues.

## 2. What Is Animal Welfare?

Harrison [9], in her book ‘Animal Machines’, spread animal welfare awareness to the British and European public by informing them that animals experience stress, pain, fear, and joy. Singer [10], in his book ‘Animal Liberation’, adds a philosophical and ethical concept to animal welfare, emphasizing two basic principles: the rejection of speciesism, and the knowledge that animals are sentient beings. Later, Hughes [11] defined animal welfare as a state of complete mental and physical health in which an animal is in harmony with its environment. In addition, Carpenter [12] defined it as animals living or adapting without suffering to the environments provided by human beings. Dawkins [13] re-suggested the need to consider both physical health and mental well-being for animal welfare, emphasizing that scientific research on subjective feelings should be supported along with welfare promotion.

The British government established the Farm Animal Welfare Council (FAWC), an independent advisory body, in 1979, and presented five freedoms (freedom from hunger and thirst; freedom from discomfort; freedom from pain, injury, or disease; freedom to express normal behavior; and freedom from fear and distress) for farm animal welfare in 1993 [14]. Since the 2000s, international organizations or associations have compiled definitions of animal welfare by integrating studies and opinions. The World Organization for Animal Health (WOAH) defines animal welfare as the ability of animals to express their natural behavior in a healthy, comfortable, well-nourished, and safe environment, and not to experience bad conditions such as pain, fear, or bullying [15]. The Royal Society for the Prevention of Cruelty to Animals (RSPCA) has stated that good welfare is providing animals with everything they need to ensure their physical and mental health and well-being [16]. In summary, animal welfare refers to the physical, mental, and emotional well-being of animals which are sentient beings that can experience pain, suffering, and distress. Furthermore, it encompasses the conditions in which animals are kept, the care they receive, and the ways in which they are treated by humans. Thus, it is of utmost importance for poultry animal caretakers and related personnels to understand the basis of animal welfare in commercial broiler production.

## 3. Assessment of Animal Welfare

Animal welfare can be evaluated through various indicators, such as animal’s health, behavior, and physiology, as well as the conditions in which they are kept and the treatment they receive. How to accurately assess the state of animal welfare at farm levels is a central issue in animal welfare research [3]. Thus, various assessment criteria to evaluate the state of animal welfare are needed to accommodate societal concerns and market demands [4].

In Europe, studies on how to evaluate animal welfare have been conducted since the early 2000s. Dawkins [3] presented a broad framework for animal welfare assessment by proposing two key questions in animal welfare evaluation: whether the animal is healthy and whether the animal has what it wants. Afterwards, detailed welfare assessment tools for livestock farms were developed and established by the Welfare Quality^®^ [5] joint project of the Welfare Quality Network, a collaboration of more than 30 institutions and scientists with support from the European Commission. The AssureWel [6] was developed by the collaborative project of the University of Bristol, the Soil Association, and the Royal Society for the Prevention of Cruelty to Animals (RSPCA). Welfare Quality^®^ broiler assessment protocol presents eighteen welfare measures under four principles (‘good feeding’, ‘good housing’, ‘good health’, and ‘appropriate behavior’), and the final result is integrated into one overall category and derived (‘excellent’, ‘enhanced’, ‘acceptable’, or ‘not classified’) [17]. The developed measures, in the majority, were derived from established research methods already described in the literature. For example, the gait scoring method of broilers was developed by Kestin et al. [18], and feather cover scoring for laying hens was improved from Hughes and Duncan [19]. As such, the animal welfare assessment indicators for broilers are, for example, stocking density, air quality such as temperature and humidity in the house, footpad dermatitis, hock burn, feather dirtiness or damage, gait score, etc., and each criterion was scored [5,6,7].

Butterworth [8] took a more systematic approach by classifying the on-farm welfare assessment methods into resource-based measures (RBM) and outcome-based measures (OBM). Resource-based measures (RBM) are based on assessing the resources available to the birds in terms of their housing, environment, and management practices, such as stocking density, litter quality, lighting and air quality, etc. These measures aimed to ensure that birds have access to the resources they need for their well-being. Outcome-based measures (OBM), also known as animal-based measures, focus on assessing the actual welfare outcomes of the birds, such as footpad dermatitis, hock burn, contamination or damage to feathers (feather score), gait score, mortality, etc. These more direct measures can be used as a warning signal and indicator of compromised animal welfare. By using a combination of both RBM and OBM, poultry farmers can obtain a more comprehensive assessment of their birds’ welfare, and provide valid information on animal welfare to consumers and commercial users [8].

## 4. Footpad Dermatitis as an Indicator for Broiler Welfare

Footpad dermatitis is a common condition that affects the footpads of broiler chickens, which are bred for meat production [20,21]. Footpad dermatitis is known by multiple names, such as pododermatitis and contact dermatitis, all of which refer to a condition that is characterized by inflammation, erosion, and ulceration of the skin on the underside of the foot [22]. This condition can cause pain and discomfort for the birds, and in severe cases, it can lead to lameness and decreased mobility [23,24]. Footpad dermatitis is recognized as an important welfare indicator for broiler chickens because it is a painful and often preventable condition that is caused by poor management practices [22,25]. Footpad dermatitis can be used as an indirect measure of welfare because it reflects the overall quality of the chicken’s environment and the level of care provided by the farmer [26].

The severity of footpad dermatitis can be assessed using a standardized scoring system, such as the Welfare Quality^®^ scoring system [5]. This system rates footpad condition on a scale of 0 to 4, with 0 indicating normal and 4 indicating severe ulceration and inflammation. Alternatively, a scale of 0 to 2 is sometimes used in the field. The scoring system takes into account the extent and severity of the lesions, as well as the degree of inflammation and the presence of infection [21,25]. Some of the main factors that can affect footpad dermatitis in broilers are discussed below.

### 4.1. Litter Quality

Litter quality is a key factor that can influence the development of footpad dermatitis in broiler chickens [1,27,28,29]. Litter that is wet and contaminated with feces or urine can soften the footpad and make it more susceptible to damage and ulceration [30,31,32,33]. Taira et al. [28] reported that with 18,500 chicks at 825 m^2^/house (stocking density of 22.4 birds/m^2^), when litter moisture increased from 30.9% to 56.5%, the footpad dermatitis score increased from an average of 0 to 2.92 points by day 42, especially noticeable from day 21. The critical ammonia concentration is unknown, but high levels of ammonia in the litter can also cause chemical burns to the footpad, leading to inflammation and ulceration [34], but other studies have not found a causal relationship [35,36]. In addition, research results have shown that the characteristics of the litter material (e.g., moisture absorption and release) have different effects on footpad dermatitis, and mortar sand and door filler have been reported to be effective in preventing footpad dermatitis [26].

### 4.2. Stocking Density

Overcrowding or high stocking densities can increase the risk of footpad dermatitis in broiler chickens, as the birds are more likely to come into contact with wet or contaminated litter and have limited space to move around [37,38,39,40]. However, studies by Granquist et al. [24] and Ekstrand et al. [32] have shown that stocking density had a low correlation with footpad dermatitis or leg health. The latter study indicates that stocking density did not affect footpad dermatitis if the critical stocking density was not reached [41,42]. Buijs et al. [42] reported that footpad dermatitis was negatively affected at densities above 56 kg/m^2^.

### 4.3. Drinking System and Feed

The drinking system may also contribute to the development of footpad dermatitis [1]. The drinking cup type is known to prevent litter moisture and the development of footpad dermatitis due to wet litter [43]. If the type of the watering system is not suitable or the water pressure is too high, the amount of water that falls onto the litter upon drinking increases the risk of litter moisture and microbial growth, which can contribute to the development of footpad dermatitis [33]. Diet can also play a role in the development of footpad dermatitis in broiler chickens. An imbalanced diet with inadequate levels of certain nutrients, such as biotin, zinc, and choline, can contribute to poor skin and footpad health, making the birds more susceptible to footpad dermatitis [44,45]. Youssef et al. [35] reported that dietary biotin (2000 µg/kg of diet) reduced the incidence of footpad dermatitis when litter had 25% moisture, but it failed to reduce it when litter had higher moisture contents (73% moisture). Abd El-Wahab et al. [46] recommended providing high Zn and biotin levels (approximately 2000 µg/kg vs. normal levels of 300 µg/kg) for broiler footpads.

### 4.4. Genetics

Genetic factors can play a role in the development of footpad dermatitis in broiler chickens [47,48]. In the experiment conducted by Kjaer et al. [47], the slow-growing dual-purpose strain did not develop footpad dermatitis over a ten-week period, while the fast-growing conventional broiler hybrid Ross 308 strain showed approximately 40% incidence within six weeks. In other words, breeding for fast growth and increased meat yield can result in a higher incidence of footpad dermatitis, as the rapid growth rate can put additional stress on the footpad and make it more susceptible to damage [49]. This is because weight during the rearing period is a factor affecting footpad dermatitis, and is an important parameter along with litter quality [50,51].

### 4.5. Climate and Ventilation

Environmental factors such as temperature, humidity, and ventilation can indirectly influence the development of footpad dermatitis [30]. High humidity levels can increase moisture in the litter, while poor ventilation can lead to high levels of ammonia and other toxic gases, both of which can contribute to the development of footpad dermatitis [21,52,53]. For example, the experiment conducted by de Jong et al. [29] found that when ventilation was reduced during winter, there was a deterioration in footpad dermatitis, leading to an increase in the proportion of score 2 by over 50%. Heat stress due to high temperatures can cause panting, reduce physical activity, and increase contact time with litter [54], and it has been reported that there is a positive correlation between heat stress and the development of footpad dermatitis [55].

## 5. Hock Burn as an Indicator for Broiler Welfare

Hock burn is a condition that occurs in broiler chickens due to prolonged contact with litter material, which leads to skin damage, inflammation, and discomfort in the hock region, and is correlated with footpad dermatitis [21,32,47]. Hock burn is considered one of the most common and significant welfare concerns in the broiler industry, and it is often used as an indicator for evaluating broiler welfare due to its prevalence and severity [1,56]. In other words, mild hock burn, which only affects the surface of the skin, is generally considered less severe than severe hock burn, which can penetrate the skin and cause tissue damage. Severe hock burn is often associated with chronic pain, discomfort, and impaired mobility, which can significantly impact the bird’s welfare [20,57]. Hock burn is often linked to poor litter management, weight gain, and can eventually increase the potential for disease outbreaks [30,58,59].

The severity of hock burn can be assessed using a standardized scoring system, such as the Welfare Quality^®^ scoring system [5], and this system grades the condition of the hock on a scale of 0–4 with 0 being normal and 4 being severe status. To reduce the incidence and severity of hock burn, broiler producers can implement various management strategies, including improving litter quality, providing adequate space and perches, and reducing stocking densities [29,60]. Some of the factors that may influence the development of hock burn in broilers are discussed below.

### 5.1. Litter Quality

As with footpad dermatitis, the litter material used in broiler housing can significantly impact the development of hock burn [27,29,61]. Wet, damp, or soiled litter can lead to increased microbial growth, which can cause skin irritation and inflammation [20,59,60]. Indeed, Haslam et al. [60] found that there was a significant positive correlation with litter quality (r = 0.475, *p* = 0.01), and a very weak correlation between hock burn and footpad dermatitis (r = 0.149, *p* = 0.07). Kristensen et al. [58] also observed a weak positive correlation between contact dermatitis on the footpad and hock (r = 0.19, *p* = 0.007), along with a positive correlation between hock burn and body weight (r = 0.48, *p* < 0.001). The reason for these results is that the factor that has a greater influence on hock burns compared to footpad dermatitis is the broiler weight during the rearing period [50,51], and there is a possibility of mild hock burns after lesions first progressed in the pads and then exacerbated footpad dermatitis [50]. Thus, it is estimated that the cause of contact dermatitis and hock burn is the same, but the contact area and the factors affecting them are different.

### 5.2. Body Weight

Broiler breed and growth rate can influence the development of hock burn [37,51]. Haslam et al. [60] reported strong positive correlations with live weight (r = 0.541, *p* = 0.01) and age (r = 0.484, *p* = 0.01) at slaughter. Fast-growing commercial broilers such as Ross 308, Cobb 500, and Arbor Acres are more susceptible to hock burn due to their increased body weight, which puts more pressure on their hock joints [50,51]. Kjaer et al. [47] observed that the slow-growing dual-purpose strains showed little hock burn over a 10-week period, whereas the fast-growing hybrid broiler strain Ross 308 showed a rapid increase in the percentage of birds affected between 4 and 6 weeks (375 (88%) of 424 birds). Heavier birds may have more difficulty standing and moving around, which can increase their exposure to litter material and exacerbate hock burn [60].

### 5.3. Stocking Density

The number of birds per unit area is related to manure output, which can affect litter quality and thus the incidence of hock burn [12,37,40,62]. Moreover, high stocking densities can lead to increased competition for resources such as food and water, allowing birds to spend more time perching or lying down, which can increase the chance of hock burn from prolonged contact with litter [63]. Indeed, van der Eijk et al. [64] reported that there was a significant difference in hock burn between stocking densities of 30 kg/m^2^ and 36 kg/m^2^. The low-stocking-density-raised broilers (30 kg/m^2^) had lower hock burn scores compared with those raised over 36 kg/m^2^. In summary, it is assumed that managing stocking density below 30 kg/m^2^ can prevent hock burn and improve welfare. On the other hand, Haslam et al. [60] found a negative correlation between stocking density and hock dermatitis.

### 5.4. Environmental Factors Such as Ventilation and Lighting

Other environmental factors, such as temperature, humidity, and light intensity affect litter quality; high temperature and humidity can increase the moisture content of the litter, which can lead to microbial growth and skin irritation [64,65]. In this respect, an effective ventilation system in the broiler house is essential to maintain proper air flow and temperature and humidity [66]. In addition, light intensity can affect leg health by altering the chicken’s physical activity. Deep et al. [67] found that leg lesions were caused by reducing broiler activity with extremely low intensity (1 lux), but Kristensen et al. [58] found no effect of light intensity (5 and 100 clux) and light source on footpad and hock dermatitis. The discordant results are likely due to differences in experimental design and low-intensity treatment.

## 6. Feather Cleanliness as an Indicator for Broiler Welfare

Feather scoring is a widely used quantitative indicator of hen welfare [50]. In laying hens, feather damage is the primary score, whereas in broilers, feather cleanliness (dirtiness) is the primary score [5]. The cleanliness of feathers is important for thermoregulation, and when feathers become wet or soiled by litter, they lose their protective properties and increase discomfort, which can negatively impact the welfare of the birds [5,27,29].

Assessing feather cleanliness involves visually inspecting the bird’s feathers and assigning them a score ranging from 0 to 3. In some cases, a scale of 0–2 is used, where scores 2 and 3 are combined in the field [68], where 0 indicates no soiling or contamination, and 3 indicates severe soiling or contamination [5,69]. Several factors can contribute to feather soiling or contamination, including the quality of the litter, the presence of feces or urine in the environment, and the management practices of the farm [70]. Below are some of the key factors that can affect feather cleanliness in broilers.

### 6.1. Litter Quality

As with contact dermatitis of the pads and hocks reviewed earlier, factors such as the quality of the litter or bedding material, the amount of moisture and humidity in the environment, and the presence of feces or urine can all contribute to soiling or contamination of feathers [27,29]. Boussaada et al. [71] revealed that the use of sawdust or wood shavings as bedding materials resulted in improved feather cleanliness compared with straw or crop residues, and there was a positive correlation between the feather cleanliness and the litter score integrated with moisture, water holding capacity, pH, and ammonia of the bedding (r = 0.39, *p* < 0.05). To maintain good feather cleanliness, it is important to ensure that the litter or bedding is clean, dry, and changed regularly, and that the environment is well-ventilated and free from excess moisture [50].

### 6.2. Stocking Density

Since there is a correlation between stocking density and litter quality, there could be a correlation between plumage cleanliness and stocking density [64,72]. Van der Eijk et al. [64] observed a reduction in feather cleanliness scores, as stocking density increased from 24 kg/m^2^ to 42 kg/m^2^. That is, the higher the stocking density, the higher the litter moisture content and fecal contamination, and the higher the possibility of being exposed to an environment in which feathers become dirty [42]. Therefore, welfare measures such as feather cleanliness derived from litter quality can be improved by reducing stocking density [64]. On the other hand, there is also a study that the larger the group size of the flock, the more negatively it affects feather cleanliness, even if the stocking density is kept constant [68]. When stressed, all the broilers may flee to one side of the barn, increasing fecal contamination on that part of the floor. Now the incidence of contamination is increased regardless of the stocking density in larger flocks. In other words, the latter study emphasized that equal stocking density at different flock sizes (i.e., from small to very large flock) could have a different impact on the welfare of commercial broiler chickens.

### 6.3. Genetics

The genetics of the broiler can play a role in feather cleanliness. Other studies have shown that fast-growing broiler breeds are often dirtier than slower-growing breeds [73,74]. In the experiment conducted by Malchow et al. [73], the fast-growing commercial meat strain (Ross 308) exhibited a relatively higher proportion of score 2 and 3 in the feather cleanliness scores compared to the slow-growing dual-purpose strain (400 Lohmann Dual). In the case of breeds bred to grow quickly, it is estimated that the metabolic rate and the amount of excretion are high, which may increase the litter contact time of high-weight chickens and the possibility that the litter contamination has spread to feathers [75,76].

### 6.4. Environmental Enrichment

There are two results regarding the effect of providing environmental enrichment such as perches and platforms in rearing facilities on feather cleanliness. Firstly, studies have shown that enrichment was not associated with feather cleanliness [69,77]. Secondly, it has been observed that the presence of a high structure platform can lead to fecal contamination of the feathers of chickens below [73,78]. Until now, there has not been much interest in the effect of enrichment on feather cleanliness [79].

## 7. Gait Score as an Indicator for Broiler Welfare

Gait scoring is used as an indicator of animal welfare in commercial broilers because it can be an early sign of musculoskeletal problems, which are common in intensively reared birds [80,81,82]. These problems can be caused by a range of factors, including genetics, body weight, and management practices, and are known to be largely influenced by genetic breeding of modern commercial broilers, which are typically selectively bred to grow rapidly [58,83,84,85]. The most common musculoskeletal problems in broiler chickens are leg disorders, such as tibial dyschondroplasia, femoral head necrosis, and osteomyelitis, which can result in poor gait scores [1,82,86].

Gait scoring is typically performed by trained observers who watch the birds walk a short distance, usually around 3–5 m, on a flat surface [5]. The gait scoring system used for broiler chickens is typically based on a 5-point scale, with a score of 0 representing a normal gait and scores of 1–5 indicating varying degrees of abnormality [18,87]. A score of 1 represents a slight defect that is difficult to define, 2 is definite and identifiable defect, but it does not hinder movement, 3 is an obvious gait defect that affects the broiler’s ability to locomotion and accelerate, 4 is a severe gait defect that means the broiler will only walk a couple of steps if driven before sitting down, and 5 indicates complete immobility [5,18]. Poor gait scores are associated with broiler welfare such as reduced mobility and access to food and water, which can all impact broiler welfare, and can result in culling in commercial rearing, resulting in economic losses for producers [88,89]. Some of the main factors that can affect the gait score of broilers are presented as follows.

### 7.1. Genetics

Over the years, broilers have been selectively bred for rapid growth and efficiency, resulting in birds with a high body weight and a large proportion of breast meat [90]. However, this selective breeding has also led to certain genetic traits that can contribute to musculoskeletal problems [91,92]. In other words, a positive correlation between fast-growing broiler breeds and walking ability was found, and correlations were also found with other outcome-based measures [74,88,93]. So, in order to improve this, research is being conducted to develop slow-growing strains to improve welfare [94].

### 7.2. Age and Weight

As broilers grow and become heavier, their bodies put increasing stress on their musculoskeletal system, which can lead to a deterioration of the gait score, including leg disorders and lameness [93,95]. Wilhelmsson et al. [93] reported that the percentage of broilers with a gait score of 2–5 increased from 8.8% in week 6 to 85% in week 9 based on an experiment conducted with 328 birds of the Ross 308 strain. Weight gain in broilers is characterized by a forward shift in the center of gravity due to rapid growth of the pectoral muscles and relatively short legs relative to body weight, which makes the bird’s gait inefficient and make it tire quickly [96].

### 7.3. Management Practices

Management practices, such as stocking density, lighting, and litter, can impact the gait score of broiler chickens [89]. Van der Eijk et al. [64] showed that the proportion corresponding to a gait score 3–5 increases as stocking density increases from 24 kg/m^2^ to 42 kg/m². High stocking densities lack room for birds to move and exercise, and may have more complex effects on locomotion, reflecting environmental loads from increased biomass such as additional ammonia and litter moisture [42]. On the other hand, some studies have reported that stocking density did not affect gait [97,98]. In addition, the dark period is beneficial for leg health and walking ability, since broilers reduce or stop feed intake during dark periods, which results in decreased growth rates [89,99]. According to the European Commission’s Broiler Directive, animal welfare standards stipulate a minimum continuous dark period of at least 6 h [100]. Additionally, as wet litter has a negative effect on walking, it is necessary to replace it with dry litter in addition to provision of adequate ventilation [29].

## 8. Conclusions

It is evident that there is a growing concern on the welfare of broilers in commercial production systems. As most broiler chickens are still raised in intensive confinement systems, alternative production systems that prioritize animal welfare and sustainability are gaining interest. In this context, it is important to identify welfare indicators for broilers and provide consumers with reliable information and evidence. As reviewed here, animal welfare is defined as the physical, mental, and emotional well-being of animals that can experience pain, suffering, and distress. Various assessment criteria have been developed to assess broiler welfare indicators and have recently been applied to evaluate their status at the farm levels [101,102]. Finally, research is needed to seek measures such as environmental enrichments or managements to improve animal welfare of broiler chickens during production.

## Data Availability

Data are contained within the article.

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
