# Peer review of "Assessment of Welfare Problems in Broilers: Focus on Musculoskeletal Problems Associated with Their Rapid Growth"

_animals, 2024, doi:10.3390/ani14071116_

Round 1

Reviewer 1 Report

Comments and Suggestions for Authors

Dear Authors,

You have organized the factors affecting animal welfare very well.

The design of the subheadings is also quite successful. But you have discussed the factors very briefly. This limits the understandability of the information.

In particular, environmental conditions such as lighting, temperature, etc. need to be written in more detail.

It would be more useful if other subheadings were given in a little more detail.

There are few references to research published in the recent five years. Numerous research have been conducted on the issue in recent years. It would be more suitable to include them also. The articles are referenced in the appropriate parts.

Why didn't you define mortality? Mortality rate is also included as an animal welfare criterion.

It appears that the effect of production systems on animal welfare has not been investigated.

4. Footpad dermatitis as an indicator for broiler welfare

It appears that the effect of production systems on foot dermatitis has not been investigated. Information regarding this factor needs to be provided.

Lines 139-146: The litter material is a factor that influences litter quality. Including research on litter materials might be beneficial.

            https://doi.org/10.1016/j.livsci.2022.105145

            http://doi.org/10.34233/jpr.1059710

Lines 148-154: More comprehensive and specific research on stocking density is required.

Lines 181-188: These factors have been insufficiently researched for their influence. Additional items are necessary.

5. Hock burn as an indicator for broiler welfare

Lines 242-243: Add the value and importance level of the relationship.

6. Feather cleanliness as an indicator for broiler welfare

Did you take into consideration breast discomfort or body feather scoring (neck, back, wing, tail) while determining the feather score? This section lacks context.

In addition, information about breast irritation should be given.

7. Gait score as an indicator for broiler welfare

Lines 335-337: It would be more appropriate to state these articles clearly.

Author Response

#1 reviewer:

Reviewer Comments in Page: Dear Authors, You have organized the factors affecting animal welfare very well. The design of the subheadings is also quite successful. But you have discussed the factors very briefly. This limits the understandability of the information. In particular, environmental conditions such as lighting, temperature, etc. need to be written in more detail. It would be more useful if other subheadings were given in a little more detail. It would be more useful if other subheadings were given in a little more detail.

Authors’ Response: We appreciate the Reviewer’s comments. Environmental conditions such as lighting and temperature have been briefly supplemented (L198-201). Subheadings were listed in order of priority that directly affect each welfare indicator, and some were refined and revised (L258).

There are few references to research published in the recent five years. Numerous research have been conducted on the issue in recent years. It would be more suitable to include them also. The articles are referenced in the appropriate parts.

Authors’ Response: Thank you for the review comments. In the next study, we will refer to recent research trends.

Why didn't you define mortality? Mortality rate is also included as an animal welfare criterion.

Authors’ Response: Mortality is also included in animal welfare criterion as part of OBMs. However, it has been detailed in other previous papers written by other researchers (Jacobs et al., 2017, https://doi.org/10.3382/ps/pew353).

It appears that the effect of production systems on animal welfare has not been investigated.

Authors’ Response: The impact of the production system on animal welfare has been written in detail, focusing on stocking density and management practices except for the Footpad dermatitis chapter.

4. Footpad dermatitis as an indicator for broiler welfare

It appears that the effect of production systems on foot dermatitis has not been investigated. Information regarding this factor needs to be provided.

Authors’ Response: It is partially described in 4.5. the climate and ventilation section, and will be reflected in the next study.

Lines 139-146: The litter material is a factor that influences litter quality. Including research on litter materials might be beneficial.

Authors’ Response: Added review of litter materials (L157-160).

Lines 148-154: More comprehensive and specific research on stocking density is required.

Authors’ Response: There is a previous review paper on stocking density by Butterworth (2019,

https://doi.org/10.1079/PAVSNNR201914039), which was not discussed in detail in this paper.

Lines 181-188: These factors have been insufficiently researched for their influence. Additional items are necessary.

Authors’ Response: We have added a review of the relationship between heat stress and footpad dermatitis (L201-204).

5. Hock burn as an indicator for broiler welfare

Lines 242-243: Add the value and importance level of the relationship.

Authors’ Response: Done as requested (L259-260).

6. Feather cleanliness as an indicator for broiler welfare

Did you take into consideration breast discomfort or body feather scoring (neck, back, wing, tail) while determining the feather score? This section lacks context. In addition, information about breast irritation should be given.

Authors’ Response: In broiler chickens, the feather score is not measured by each part of the chicken, but rather evaluates the cleanliness (dirtiness) of the overall feather.

7. Gait score as an indicator for broiler welfare

Lines 335-337: It would be more appropriate to state these articles clearly.

Authors’ Response: Thank you for the comment.

Reviewer 2 Report

Comments and Suggestions for Authors

The welfare of broilers is of growing concern. The authors discuss in detail current concepts of animal welfare and the components that are considered relevant to animal welfare. One of these aspects is health. Due to their rapid weight gain, there is a very high probability that a number of serious abnormalities and damage will occur in the musculoskeletal system of broiler chickens. The authors focus on three problems associated with these growth-induced problems: "Hock burn", "Feather cleanness" and "Gait scores". These are considered as indicators of animal welfare in broilers. These indicators are, directly or indirectly, the result of the enormous weight gain, whereby the broiler chicken experiences more and more restrictions in its mobility. Contact with the (often heavily soiled floor litter) leads to the problems mentioned above. Other factors, genetic and environmental, as well as stocking density, also contribute to these problems.

The musculoskeletal problems are very pronounced in broilers and probably compromise animal welfare the most in these animals. However, animal welfare encompasses many more aspects, of which emotional status is seen as a key component in almost all definitions of animal welfare. None of these aspects are discussed further in the manuscript. It therefore makes sense to state the focus of the manuscript more clearly in the introduction. The introduction should therefore also be rewritten in a less general and more detailed way. A strong shortening and better focussing would increase the readability of the manuscript and would do more justice to the topics discussed.

To further improve the readability of the manuscript, authors should shorten long sentences or divide them into several shorter sentences.

Specific comments

Line 2: I suggest changing the title: “Assessment of welfare problems in broilers: focus on musculoskeletal  problems associated with their rapid growth.” The authors will certainly be able to formulate an even better title

Line 17: in the commercial broiler farms.

Lines 28-29: It is a bold claim that OBM measures can help identify ANY welfare problem.

Lines 43-44: I'm sorry, I don't understand this sentence.

Lines 107-108: What is meant by "and each criterion was scored"?

Line 108: methods

Line 120 ff.: Any disease or injury affects the health of an animal and will affect its welfare. However, welfare is NOT the absence of disease or injury. Animal welfare is a complex construct that cannot be reduced to a single condition.

Lines 132-134: This sentence is unclear and needs to be reworded.

Line 153: Which study? At the end of this sentence, the authors refer to two studies.

Lines 158-159: Rephrase the sentence: The drinking cup type is known to prevent litter moisture and the development of footpad dermatitis due to wet litter.

Lines 167 -168: What exactly do 25% moisture and 73% moisture mean?

Lines 206-207: Some of the factors that may influence the development of hock burn in  broilers are discussed below.

Lines 212 ff.:  I disagree that this is a fairly strong correlation. There are many suggestions on how to qualify the strength of a correlation, e.g.":

Absolute value of r versus strength of the  relationship:

r < 0.25                 No relationship

0.25 < r < 0.5      Weak relationship

0.5 < r < 0.75      Moderate relationship

r > 0.75                 Strong relationship.

See, e.g., DOI: 10.1213/ANE.0000000000002864

Authors may find more publications dealing with the interpretation of correlations.

Lines 242-243: There are conflicting results. How can increasing stocking density be detrimental in some cases and beneficial in others?

Lines 253-255:  Explain better.

Lines 262-265: Explain better. See also my comments to lines 132-134.

Line 276: "integrated with moisture, water holding capacity, pH and ammonia of the bedding".

Line 281: Rephrase: correlation does NOT in itself indicate causation.

Lines 289-291: I don't know if this explanation has been discussed in the scientific literature: When stressed, all the broilers may flee to one side of the barn, increasing faecal contamination on that part of the floor. Now the incidence of contamination is increased regardless of the stocking density (which assumes that all broilers are evenly distributed over the entire shed floor).

Line 315: Replace “improved” with “selectively bred”

Line 482: Change to DOI:202093/ps/85.8.1342

Author Response

#2 reviewer:

Reviewer Comments in Page: he welfare of broilers is of growing concern. The authors discuss in detail current concepts of animal welfare and the components that are considered relevant to animal welfare. One of these aspects is health. Due to their rapid weight gain, there is a very high probability that a number of serious abnormalities and damage will occur in the musculoskeletal system of broiler chickens. The authors focus on three problems associated with these growth-induced problems: "Hock burn", "Feather cleanness" and "Gait scores". These are considered as indicators of animal welfare in broilers. These indicators are, directly or indirectly, the result of the enormous weight gain, whereby the broiler chicken experiences more and more restrictions in its mobility. Contact with the (often heavily soiled floor litter) leads to the problems mentioned above. Other factors, genetic and environmental, as well as stocking density, also contribute to these problems.

The musculoskeletal problems are very pronounced in broilers and probably compromise animal welfare the most in these animals. However, animal welfare encompasses many more aspects, of which emotional status is seen as a key component in almost all definitions of animal welfare. None of these aspects are discussed further in the manuscript. It therefore makes sense to state the focus of the manuscript more clearly in the introduction. The introduction should therefore also be rewritten in a less general and more detailed way. A strong shortening and better focussing would increase the readability of the manuscript and would do more justice to the topics discussed.

Authors’ Response: We appreciate the Reviewer’s comments and suggestions. Following the Reviewer’s advice, we have revised the introduction to clearly focus on outcome-based measures.

To further improve the readability of the manuscript, authors should shorten long sentences or divide them into several shorter sentences.

Authors’ Response: To reduce the length of some sentences, I divided them into several shorter sentences. However, the research content of the referenced literature can be expressed in one sentence.

Specific comments

Line 2: I suggest changing the title: “Assessment of welfare problems in broilers: focus on musculoskeletal  problems associated with their rapid growth.” The authors will certainly be able to formulate an even better title

Authors’ Response: Done as requested (L2-3).

Line 17: in the commercial broiler farms.

Authors’ Response: Done as requested (L18).

Lines 28-29: It is a bold claim that OBM measures can help identify ANY welfare problem.

Authors’ Response: Completed (L30).

Lines 43-44: I'm sorry, I don't understand this sentence.

Authors’ Response: We paraphrased the sentence (L45-48).

Lines 107-108: What is meant by "and each criterion was scored"?

Authors’ Response: This means that each criterion is scored and can be scored by the evaluator.

Line 108: methods

Authors’ Response: Completed (L117).

Line 120 ff.: Any disease or injury affects the health of an animal and will affect its welfare. However, welfare is NOT the absence of disease or injury. Animal welfare is a complex construct that cannot be reduced to a single condition.

Authors’ Response: Thank you for the comment. The complex meaning of animal welfare was mentioned in the history and definition of welfare section in Chapter 2.

Lines 132-134: This sentence is unclear and needs to be reworded.

Authors’ Response: The sentence was divided and rewritten into two sentences (L143-145).

Line 153: Which study? At the end of this sentence, the authors refer to two studies.

Authors’ Response: The sentence was revised to indicate the researcher (L18).

Lines 158-159: Rephrase the sentence: The drinking cup type is known to prevent litter moisture and the development of footpad dermatitis due to wet litter.

Authors’ Response: Done as requested (L173-174).

Lines 167 -168: What exactly do 25% moisture and 73% moisture mean?

Authors’ Response: These were the moisture content of the dry and wet litter treatments measured in the experiment conducted by Youssef et al. (2011).

Lines 206-207: Some of the factors that may influence the development of hock burn in broilers are discussed below.

Authors’ Response: Done as requested (L225-226).

Lines 212 ff.:  I disagree that this is a fairly strong correlation. There are many suggestions on how to qualify the strength of a correlation, e.g.":

Absolute value of r versus strength of the relationship:

r < 0.25                 No relationship

0.25 < r < 0.5      Weak relationship

0.5 < r < 0.75      Moderate relationship

r > 0.75                 Strong relationship.

See, e.g., DOI: 10.1213/ANE.0000000000002864

Authors may find more publications dealing with the interpretation of correlations.

Authors’ Response: Completed (L231).

Lines 242-243: There are conflicting results. How can increasing stocking density be detrimental in some cases and beneficial in others?

Authors’ Response: Haslam et al.'s (2007) results may seem counterintuitive on the surface because higher stocking densities are more likely to reduce litter quality and thus increase contact dermatitis. However, their discussion points out that the influence of bird age and/or body weight could have a much greater impact on hock burn incidence than litter quality.

Lines 253-255:  Explain better.

Authors’ Response: Done as requested (L273-275).

Lines 262-265: Explain better. See also my comments to lines 132-134.

Authors’ Response: The sentence was divided and rewritten into two sentences (L284-286).

Line 276: "integrated with moisture, water holding capacity, pH and ammonia of the bedding".

Authors’ Response: Done as requested (L298-299).

Line 281: Rephrase: correlation does NOT in itself indicate causation.

Authors’ Response: Revised the sentence to show only correlations (L273-275).

Lines 289-291: I don't know if this explanation has been discussed in the scientific literature: When stressed, all the broilers may flee to one side of the barn, increasing faecal contamination on that part of the floor. Now the incidence of contamination is increased regardless of the stocking density (which assumes that all broilers are evenly distributed over the entire shed floor).

Authors’ Response: Done as requested (L298-299).

Line 315: Replace “improved” with “selectively bred”

Authors’ Response: Done as requested (L298-299).

Line 482: Change to DOI:202093/ps/85.8.1342

Authors’ Response: The existing DOI is correct.

Reviewer 3 Report

Comments and Suggestions for Authors

1.          This review article discusses the measurable items and their influencing factors for assessing animal welfare in broiler chicken.

2.          In the discussion of the development of animal welfare, a very important link is missing, that is, the contribution of FAWC and the five-freedoms it established.

3.          This article mentions RBM and OBM, but the discussion only focuses on OBM indicators, and only discusses footpad dermatitis, hock burn, feather conditions and gait score. Most of their influencing factors are similar. Management is only discussed in gait. In fact, management is related to all other indicators.

4.          In addition, the relationship between humans and animals, behavioral observations, and physiological indicators have not been discussed. These also belong to OBM.

Author Response

#3 reviewer:

Reviewer Comments in Page: 1. This review article discusses the measurable items and their influencing factors for assessing animal welfare in broiler chicken.

2. In the discussion of the development of animal welfare, a very important link is missing, that is, the contribution of FAWC and the five-freedoms it established.

Authors’ Response: Sentences about FAWC and the five-freedoms have been added to Chapter 2 (L72-76).

3. This article mentions RBM and OBM, but the discussion only focuses on OBM indicators, and only discusses footpad dermatitis, hock burn, feather conditions and gait score. Most of their influencing factors are similar. Management is only discussed in gait. In fact, management is related to all other indicators.

Authors’ Response: We appreciate the Reviewer’s comments. The introduction has been revised to specify the purpose and direction of this review. The context has been modified so that it is natural to discuss OBM. However, we will take the comments into account in the next study to ensure it is a comprehensive review of broiler welfare.

4. In addition, the relationship between humans and animals, behavioral observations, and physiological indicators have not been discussed. These also belong to OBM.

Authors’ Response: Thank you for the comment. We will refer to this in our next research.

Round 2

Reviewer 1 Report

Comments and Suggestions for Authors

Dear Authors,

The indicated revisions have been performed.

Author Response

#1 reviewer: The indicated revisions have been performed.

Authors’ Response: Thank you for the comment.

Reviewer 2 Report

Comments and Suggestions for Authors

The readability of the manuscript has been improved and the scope of this review is now much clearer than in the original submission. I suggest that three sentences be rephrased before the paper is ready for publication (see comments in PDF file).

Author Response

#2 reviewer: The readability of the manuscript has been improved and the scope of this review is now much clearer than in the original submission. I suggest that three sentences be rephrased before the paper is ready for publication (see comments in PDF file).

Authors’ Response: Per reviewer’ suggestion, we have re-written the sentences (L124-125), (L248-251), (L278-280).